# Clinical Outcome of Low-Grade Myofibroblastic Sarcoma in Japan: A Multicenter Study from the Japanese Musculoskeletal Oncology Group

**DOI:** 10.3390/cancers15082314

**Published:** 2023-04-15

**Authors:** Munehisa Kito, Keisuke Ae, Masanori Okamoto, Makoto Endo, Kunihiro Ikuta, Akihiko Takeuchi, Naohiro Yasuda, Taketoshi Yasuda, Yoshinori Imura, Takeshi Morii, Kazutaka Kikuta, Teruya Kawamoto, Yutaka Nezu, Ichiro Baba, Shusa Ohshika, Takeshi Uehara, Takafumi Ueda, Jun Takahashi, Hirotaka Kawano

**Affiliations:** 1Department of Orthopaedic Surgery, Shinshu University School of Medicine, 3-1-1 Asahi, Matsumoto 390-8621, Japan; 2Department of Orthopaedic Surgery, Cancer Institute Hospital, Japanese Foundation for Cancer Research, 3-8-31 Ariake, Koto-ku, Tokyo 135-8550, Japan; 3Department of Orthopaedic Surgery, Graduate School of Medical Sciences, Kyushu University, 3-1-1 Maidashi, Higashi-ku, Fukuoka 812-8582, Japan; 4Department of Orthopaedic Surgery, Nagoya University Graduate School of Medicine, 65 Tsurumai, Showa-ku, Nagoya 466-8560, Japan; 5Department of Orthopaedic Surgery, Graduate School of Medical Sciences, Kanazawa University, 13-1 Takaramachi, Kanazawa 920-8641, Japan; 6Department of Orthopaedic Surgery, Osaka University Graduate School of Medicine, 2-2 Yamadaoka, Suita 565-0871, Japan; 7Department of Orthopaedic Surgery, National Hospital Organization Osaka National Hospital, 2-1-14 Houenzaka, Chuo-ku, Osaka 540-0006, Japan; 8Department of Orthopaedic Surgery, University of Toyama, 2630 Sugitani, Toyama 930-0194, Japan; 9Department of Orthopaedic Surgery, Osaka International Cancer Institute, 3-1-69 Otemae, Chuo-ku, Osaka 540-0008, Japan; 10Department of Orthopaedic Surgery, Kyorin University Faculty of Medicine, 6-20-2 Shinkawa, Tokyo 181-8621, Japan; 11Department of Musculoskeletal Oncology and Orthopaedic Surgery, Tochigi Cancer Center, 4-9-13 Yonan, Utsunomiya 320-0834, Japan; 12Department of Orthopaedic Surgery, Kobe University Graduate School of Medicine, 7-5-1 Kusunoki-cho, Chuo-ku, Kobe 650-0017, Japan; 13Department of Musculoskeletal Oncology and Rehabilitation, National Cancer Center Hospital, 5-1-1 Tsukigi, Chuo-ku, Tokyo 104-0045, Japan; 14Department of Orthopaedic Surgery, Osaka Medical and Pharmaceutical University, 2-7 Daigakumachi, Takatsuki 569-8686, Japan; 15Department of Orthopaedic Surgery, Hirosaki University Graduate School of Medicine, 5 Zaifu-cho, Hirosaki 036-8562, Japan; 16Department of Laboratory Medicine, Shinshu University School of Medicine, 3-1-1 Asahi, Matsumoto 390-8621, Japan; 17Department of Orthopaedic Surgery, Kodama Hospital, 1-3-2 Gotenyama, Takarazuka 665-0841, Japan; 18Department of Orthopaedic Surgery, Teikyo University School of Medicine, 2-11-1 Kaga, Itabashi-ku, Tokyo 173-0806, Japan

**Keywords:** low-grade myofibroblastic sarcoma, rare sarcoma, wide excision, radiotherapy, local relapse, prognosis

## Abstract

**Simple Summary:**

Low-grade myofibroblastic sarcoma (LGMS) is one of the rarest sarcomas. We aimed to clarify the clinical outcomes of patients with LGMS. Twenty-two patients underwent surgical treatment for the primary tumor and two underwent radical radiotherapy (RT). The best overall response in the two patients who underwent radical RT was one complete response and one partial response. Local relapse-free survival was 91.3% at 2 years and 75.4% at 5 years. Relapsed tumors were treated with surgery in two cases and radical RT in three cases. None of the patients experienced a second local relapse. Disease-specific survival was 100% at 5 years. Wide excision is recommended due to its tendency to local relapse. However, RT was considered a viable option in unresectable cases or in cases where surgery may cause significant functional impairment.

**Abstract:**

This retrospective multicenter study aimed to analyze the clinical features and prognosis of 24 patients diagnosed with LGMS between 2002 and 2019 in the Japanese sarcoma network. Twenty-two cases were surgically treated and two cases were treated with radical radiotherapy (RT). The pathological margin was R0 in 14 cases, R1 in 7 cases, and R2 in 1 case. The best overall response in the two patients who underwent radical RT was one complete response and one partial response. Local relapse occurred in 20.8% of patients. Local relapse-free survival (LRFS) was 91.3% at 2 years and 75.4% at 5 years. In univariate analysis, tumors of 5 cm or more were significantly more likely to cause local relapse (*p* < 0.01). In terms of the treatment of relapsed tumors, surgery was performed in two cases and radical RT was performed in three cases. None of the patients experienced a second local relapse. Disease-specific survival was 100% at 5 years. A wide excision aimed at the microscopically R0 margin is considered the standard treatment for LGMS. However, RT may be a viable option in unresectable cases or in cases where surgery is expected to cause significant functional impairment.

## 1. Introduction

Low-grade myofibroblastic sarcoma (LGMS) is a rare mesenchymal spindle cell neoplasm that exhibits fibromatosis-like features and differentiation of fibroblasts into myofibroblasts [1]. The disease was first described by Mentzel et al. in 1998 and classified as a novel disease in the World Health Organization (WHO) classification of tumors of soft tissue and bone in 2002 [2]. It is known to occur frequently in the subcutaneous and deep soft tissues of the head, neck, and extremities with a diffusely infiltrative growth pattern and a tendency to cause local relapse. However, because it is a low-grade malignant tumor, distant metastasis is rare. Four small retrospective observational studies [1,3,4,5], two analytical studies using the Surveillance, Epidemiology, and End Results (SEER) database [6,7], and multiple case reports [8,9,10,11,12,13,14,15,16,17,18,19,20,21,22] have been reported to date. Because the local relapse rate is higher with simple excision than with wide excision, wide excision is generally the recommended treatment method. In terms of adjuvant radiotherapy (RT)/chemotherapy, Mentzel et al. [1] and Montgomery et al. [3] reported that local relapse was not observed in patients who underwent excision and adjuvant RT; however, Xu et al. [7] recommended against the routine use of adjuvant RT/chemotherapy due to the limited effects on survival. There are few reports on radical RT in unresectable cases. Distant metastasis is rare, but the treatment strategy for distant metastasis remains controversial. Regarding survival rates, an analysis of the SEER database by Chan et al. [6] showed an 80% survival at 3 years and 76.3% at 5 years, while Xu et al. [7] reported a 93% survival at 1 year, 79% at 5 years, and 76% at 10 years. However, due to the rarity of LGMS, many aspects of the disease remain unknown, such as the survival rate. There is presently no consensus regarding the optimal treatment strategy for LGMS. The purpose of this study is to investigate treatment of LGMS cases at facilities that are members of the Japanese Musculoskeletal Oncology Group (JMOG) and to clarify the clinical outcome for LGMS.

## 2. Materials and Methods

### 2.1. Study Design and Evaluation

From 2002, when LGMS was included in the WHO classification, to 2019, patients diagnosed and treated for LGMS at 14 JMOG tertiary referral centers for musculoskeletal tumors were included in this research. This study was a retrospective, multicenter study that was approved by the institutional review board at each institution and conducted in accordance with the Declaration of Helsinki. The following clinical information was collected: age, sex, initial presentation (primary or relapse), primary site of occurrence, localization, maximum tumor diameter, treatment method for the primary tumor, complications associated with treatment, presence of local relapse, presence of distant metastases, treatment methods for local relapse/distant metastases, and oncological outcome at final follow-up. Information on the pathological margins was collected from those who underwent surgery. The pathological margins were assessed by residual (R) tumor classification [23], with R0 corresponding to no residual tumor, R1 to microscopic residual tumor, and R2 to macroscopic residual tumor. In patients who underwent RT/chemotherapy, evaluation of the response to the treatment was determined according to the RECIST guideline [24], and local relapse was defined as tumor regrowth from the time of the best overall response. The following histopathological information was collected: mitotic count, tumor necrosis, and immunohistochemical study (α-SMA, desmin, calponin, h-caldesmon, CD34, β-catenin, S-100, CK AE/AE3, and Ki-67). The mitotic count and tumor necrosis were assessed using the National Cancer Institute and French Federation of Cancer Centers Sarcoma Group (FNCLCC) grading system [25]. The pathological diagnosis was performed by experienced pathologists specializing in bone and soft-tissue tumors at each facility.

### 2.2. Statistical Analysis

The local relapse-free survival (LRFS) and disease-specific survival (DSS) were calculated using the Kaplan–Meier method. To identify factors associated with LRFS, univariate analysis was performed using the log-rank test. A *p*-value of 0.05 or less was considered statistically significant. IBM SPSS Statistics (version 28) was used for data analysis.

## 3. Results

### 3.1. Patient Characteristics

Ten males and fourteen females were included in this study. Detailed patient characteristics are shown in Table 1. The mean age was 45.7 years (11–83 years), and the condition of the sarcoma at the initial visit was primary in 23 cases and relapse in 1 case. None of the patients had distant metastases. The primary tumor sites were 10 cases in the trunk (buttock, 3 cases; back, 3 cases; chest wall, 1 case; abdominal wall, 1 case; axilla, 1 case; groin, 1 case), 5 cases in the head and neck (neck, 3 cases; vocal cord, 1 case; tongue, 1 case), 6 cases in the lower extremity (thigh, 3 cases; lower leg, 2 cases; foot, 1 case), and 3 cases in the upper extremity (upper arms, 2 cases; forearm, 1 case). In terms of localization, 13 cases were deep-seated, and 11 cases were superficial. The mean maximum tumor diameter was 4.7 cm (1–14 cm), with 15 cases < 5 cm and 9 cases ≥ 5 cm. The mean follow-up period (from the date of intervention to last follow-up) was 79 months (4–181 months).

### 3.2. Pathological Features

Histologically, spindle cells with mild atypia had proliferated while forming complex and irregular fascicles, thus indicating infiltration into the surrounding tissues (Figure 1a,b). The mitotic count score was one in all cases. There were no cases with tumor necrosis. The results of the immunohistochemical study were as follows: α-SMA, 24 positive and 0 negative; desmin, 3 positive, 20 negative, and 1 untested; calponin, 7 positive, 1 negative, and 16 untested; h-caldesmon: 0 positive, 12 negative, and 12 untested; CD34, 3 positive, 17 negative, and 4 untested; β-catenin, 0 positive, 11 negative, and 13 untested; S-100, 2 positive, 19 negative, and 3 untested; CK AE/AE3, 1 positive, 12 negative, and 11 untested; and Ki-67, mean 9.7% (20 cases, 2–40%). A-SMA, a marker for unstriated muscle cells and myofibroblasts, was positive in all cases. As for other muscle markers, desmin was negative in 20 out of 23 cases, calponin was positive in 7 out of 8 cases, and h-caldesmon was negative in all 12 cases (Figure 1c–f).

### 3.3. Treatment and Outcome of Initial Tumor

Details are shown in Table 2. Twenty-two cases underwent surgical treatment. The surgical method was wide excision in 17 cases, marginal excision in 4 cases, and intralesional excision in 1 case. One case of intralesional excision of a tumor arising from the subcutaneous tissue was performed under local anesthesia; however, this was an unplanned excision that left gross residual tumor tissues due to adhesion to muscles. The pathological margin was R0 in 14 cases, R1 in 7 cases, and R2 in 1 case. Although the indication for adjuvant RT was determined by each institution, four patients received postoperative adjuvant RT, of which three cases had an R1 pathological margin (No. 10, 11, and 18) and one case had an R0 margin in close proximity to a tumor (No. 15). Two cases (No. 23 and 24) were determined to be difficult to undergo surgical treatment and underwent radical RT (intensity-modulated radiation therapy: IMRT). The best overall response after RT was one case of complete response (Figure 2) and one case of partial response.

Local relapse occurred in 20.8% of cases (5 cases: No. 6, 15, 17, 20, 24). The mean time to local relapse was 27 months (6–36 months). The LRFS was 91.3% at 2 years and 75.4% at 5 years (Figure 3).

In univariate analysis, the only factor associated with the LRFS was the maximum tumor diameter, and tumors with a diameter of 5 cm or more were significantly more likely to exhibit local relapse (*p* < 0.01) (Table 3).

### 3.4. Treatment and Outcome of Relapsed Tumor

Details on the treatment and outcomes for relapsed tumors are shown in Table 4. Two cases underwent surgical treatment. Radical RT was performed on three patients for whom surgical treatment was determined to be difficult. Case No. 24 underwent radical RT during the initial treatment. Although the evaluation of the response was determined to be a partial response, there was tumor relapse outside the irradiated field and re-irradiation was, subsequently, performed. The type of radiation was X-ray in two cases and carbon ion in one case. None of the patients experienced a second local relapse at a mean of 79 months (22–175 months) after retreatment. A case arising from the axilla treated with carbon ion RT (No. 17) had grade 3 neuropathy according to the Common Terminology Criteria for Adverse Events version 5.0 (CTCAE). No adverse events occurred in the re-irradiated case (No. 24) at 43 months after re-irradiation.

### 3.5. Distant Metastases and Disease-Specific Survival

Distant metastases occurred in 4.2% of cases (one case: No. 20). The time to distant metastases was 36 months. The metastatic organs were the lungs and lymph nodes. The patient is currently undergoing routine follow-up without treatment, but is still alive at 22 months from the diagnosis of metastasis. The DSS was 100% at 5 years. The oncological outcome was completely disease-free in 17 cases, no evidence of disease in 3 cases, and alive with disease in 4 cases.

## 4. Discussion

LGMS was first reported by Mentzel et al. in a case series of 18 cases in 1998. Histologically, the sarcoma shows a diffusely infiltrative growth pattern and is composed of spindle-shaped tumor cells arranged in fascicles. In immunohistochemical studies of muscle markers, α-SMA can be positive in most cases, but desmin can be both positive and negative [2,26]. In addition, calponin can often be positive, but h-caldesmon is negative. This is important for its differentiation from leiomyosarcoma (which is positive for both calponin and h-caldesmon) [27]. CK/CD34/S100 is often negative. In this study, α-SMA was positive in all cases, and desmin was almost always negative. In terms of calponin and h-caldesmon, approximately half of the cases were tested, but most of them were shown to be calponin-positive and h-caldesmon-negative. The results of the immunohistochemical study were considered consistent with past reports. Mentzel et al. [1] also reported that tumor cells showed moderate nuclear atypia and the mitotic cell count was 1 to 6 figures per 10 HPF in most cases; however, no tumor necrosis was observed. Montgomery et al. [3] reported 15 cases of which 5 cases were histological grade 2 (the presence of necrosis (up to 15%) and 6 or more mitotic figures per 10 HPF). Meng et al. [4] reported 20 cases that included 6 grade 2 cases. Using the SEER database, Xu et al. [7] reported that 22.9% of cases were histologically grade 2 or 3, according to the FNCLCC grading system. These reports included non-low-grade myofibroblastic sarcomas, which might affect the survival analysis. In this study population, the mitosis count score was one for all cases, and there were no cases with tumor necrosis; thus, we believe this study contains the true number of LGMS cases.

Local relapse of LGMS has been reported to range from 13.3% to 44.4% [1,3,4,5]. The local relapse rate is especially prominent in simple excisions and accounts for up to 28.6–100% of cases. In this study, the local relapse rate was 20%. Despite 14 patients achieving R0 margins, local relapse occurred in 2 patients (14.2%). Fujiwara et al. [28] reported a 100% local control in low-grade soft-tissue sarcoma with excision margins of ≥2 mm. LGMS is, thus, believed to be a tumor with a high relapse rate among low-grade sarcomas. Although dependent on the site of occurrence, a wide excision with the preservation of as much normal tissue as possible is desirable for local treatment, rather than a simple R0 excision. Only the tumor size was shown to be statistically associated with local relapse in this study. Because local relapse of soft-tissue sarcoma is reported to be less common when the tumor diameter is less than 5 cm [29], we believe that our results are acceptable.

The use of adjuvant RT/chemotherapy for LGMS remains debatable, considering that previous reports have only included a limited number of cases. Mentzel et al. [1] reported that two patients who underwent excision and RT showed no local relapse, and Montgomery et al. [3] also reported two patients who underwent excision and RT without local relapse. Meng et al. [4] reported local relapse in one of two patients who underwent excision and RT, while local relapse occurred in three of four patients with excision and chemotherapy. Khosla et al. [14] reported no local relapse for 14 months after partial excision and RT for LGMS arising from the larynx. Peng et al. [10] reported that excision and chemotherapy for LGMS arising from the pancreas had remained continuously disease-free for 5 years. A report by Xu et al. [7] using the SEER database stated that RT and chemotherapy should not be routinely performed when a negative margin can be secured, due to the limited improvement in survival rates. In this study, postoperative RT was performed in one case with R1 marginal excision, two cases with R1 wide excision, and one case with R0 wide excision of a tumor in close proximity to the stump. Local control was obtained in 75% of cases (three cases). Therefore, it is necessary to consider more cases in a future study; however, in patients with positive excision margins, adjuvant RT may be considered. Although the recommendation of adjuvant chemotherapy may be outside the scope of this study, due to the lack of applicable patients in this study, further collection of data is warranted for a future study, considering the scarcity of previous reports on this subject matter.

Presently, there are almost no reports on the results of radical RT. Xu et al. [7] reported three patients who underwent RT without surgical treatment, but did not provide detailed outcomes. Zoltán et al. [9] performed 66 Gy radical RT for local relapse of LGMS of the tongue and reported good local control at 50 months after irradiation. On the other hand, Yu et al. [30] reported that 66 Gy of radical RT was performed on LGMS of the mandibular canal, but the tumor again increased in size 6 months later. Oral intake subsequently became impossible, leading to death from disease. In this study, radical RT (radiation dose: 60 Gy, four cases; 70.4 Gy, one case) was performed in two cases of unresectable primary tumors and three cases of relapsed tumors. In one patient who underwent irradiation of a primary tumor, re-irradiation was performed due to a local relapse that occurred outside the irradiated field. Local control was achieved 43 months after re-irradiation. In other cases, good local control was obtained after one irradiation. Although we cannot make a definitive statement due to the small number of cases, RT may be a useful treatment option in unresectable cases or in cases where surgery results in significant functional impairment due to the potential radiosensitivity of LGMS. However, one case relapsed outside of the area of irradiation. Considering that LGMS shows a growth pattern that infiltrates the surrounding tissue, close attention must be paid in setting the area for irradiation. Since the tumor is surrounded by important tissues in head and neck lesions, the use of an intensity modulated technique to reduce irradiation to normal tissues and to reduce adverse events [31], and the use of carbon ion beams and proton beams to obtain good local control with fewer adverse events [32,33,34], may be necessary in these cases. In this study, distant metastases occurred in 4.2% of patients, which was comparable to previous reports (0–9.1%) [1,3,5]. However, there is no consensus on the optimal treatment strategy following the occurrence of metastases. Because cases with metastasis in this study did not undergo post-metastatic treatment, a recommendation for treatment is outside the scope of this study.

Table 5 compares the local relapse and distant metastases rates, according to the local treatment used in previous reports and this study.

In terms of DSS, Chan et al. [6] reported 80% at 3 years and 76.3% at 5 years, and Xu et al. [7] reported 93% at 1 year, 85% at 3 years, 79% at 5 years, and 76% at 10 years. These reports are analytical studies using the SEER database. In the report by Xu et al., 22.9% of cases were histological grade 2 or 3, and 8.3% had distant metastases at initial presentation. Non-low-grade myofibroblastic sarcoma was included in their study, as evidenced by their database search term “third-edition ICD-O (ICD-O-3) histological code 8825/3: Myofibroblastoma, malignant.” In this study, DSS was 100% at 5 years, showing characteristics consistent with a low-grade tumor. Cases with a low mitotic count and no tumor necrosis were collected in this study, and we believe that the prognosis of LGMS was accurately evaluated.

There were notable limitations to this research. Firstly, this was a retrospective asymmetric observational study with a small number of patients, and the possibility of selection bias in the patients and treatments cannot be ruled out. Moreover, the sample size was too small for statistical analysis. Therefore, it is possible that the prognosis and optimal treatment cannot be accurately determined. Secondly, the examination for confirming the diagnosis was inconsistent due to the lack of a central pathology review. This carries the potential risk of diagnostic problems. However, since pathologists for tertiary referral centers for musculoskeletal tumor performed the diagnosis, we expect that most cases were correctly diagnosed. A future prospective study may warrant a larger sample size.

## 5. Conclusions

Although a wide excision aiming at the R0 margin is considered the standard treatment for LGMS, RT may provide some degree of response and may be a viable option in unresectable cases, or in cases where surgery is expected to cause significant functional impairment.

## Figures and Tables

**Figure 1 cancers-15-02314-f001:**
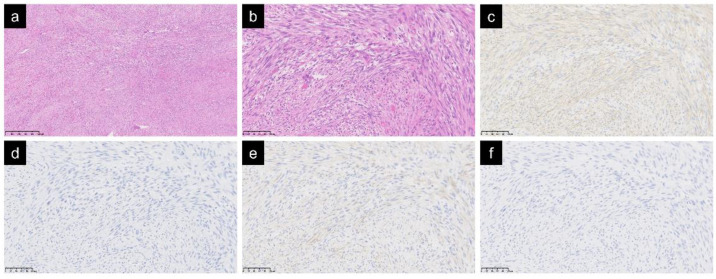
Spindle cells with mild atypia were observed with irregular fascicles. Hematoxylin and eosin staining: magnification ×50 (**a**); magnification ×200 (**b**). In an immunochemical study of muscle markers, α-SMA was positive (**c**), desmin was negative (**d**), calponin was positive (**e**), and h-caldesmon was negative (**f**); ((**c**–**f**) magnification ×200).

**Figure 2 cancers-15-02314-f002:**
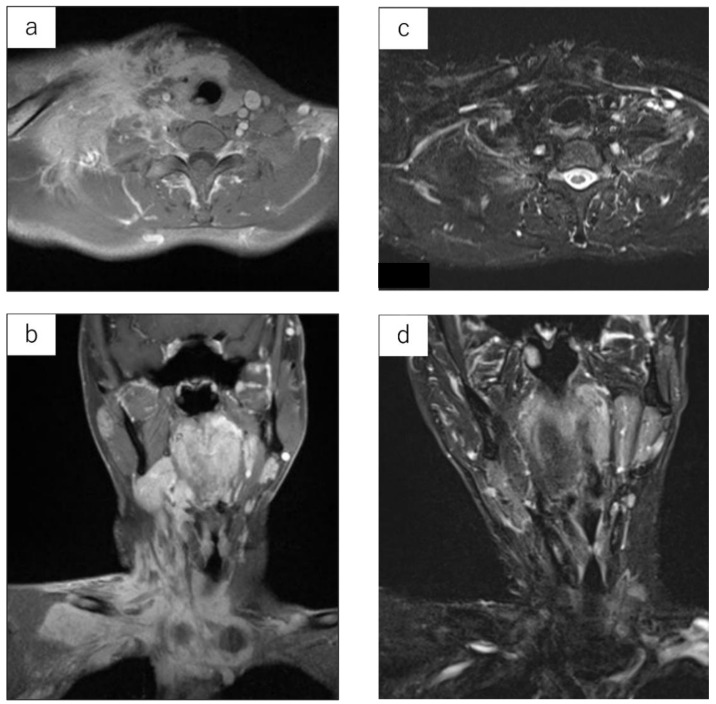
LGMS arising from the neck of a 29-year-old male. Contrast-enhanced MRI prior to irradiation. Signal changes are observed across a large area of the neck (**a**,**b**). MRI STIR image at 10.1 years after irradiation (IMRT: 60 Gy). The tumor has completely subsided (**c**,**d**).

**Figure 3 cancers-15-02314-f003:**
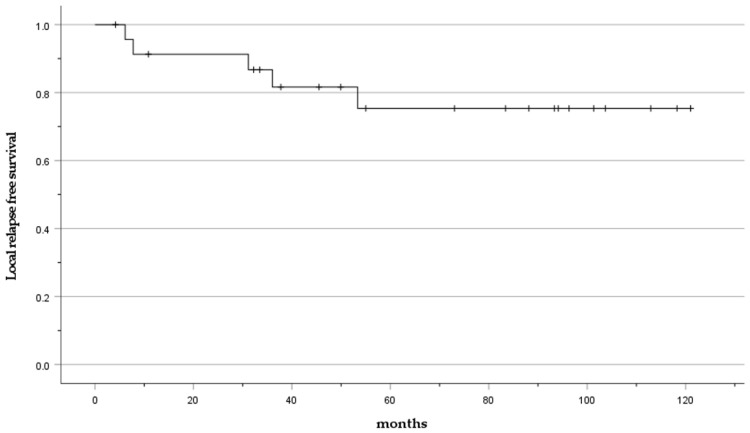
Kaplan–Meier curve presenting the LRFS.

**Table 1 cancers-15-02314-t001:** Patient characteristics.

No.	Age (y)	Sex	Initial Presentation	Primary Site	Localization	Maximum Diameter (cm)	F/U Periods (m)
1	46	F	Primary	Back	Deep	2.5	113
2	58	F	Primary	Neck	Deep	3.5	101
3	48	M	Primary	Abdominal wall	Superficial	1	104
4	64	M	Primary	Vocal cords	Deep	1.4	34
5	30	F	Primary	LE (ankle)	Superficial	1.6	83
6	67	M	Primary	Chest wall	Deep	3	161
7	11	M	Primary	UE (upper arm)	Superficial	2.8	96
8	19	F	Primary	LE (thigh)	Superficial	3.7	94
9	12	M	Primary	LE (foot)	Superficial	2.8	118
10	36	M	Primary	Back	Deep	6.1	46
11	79	F	Primary	UE (forearm)	Superficial	4	50
12	27	F	Relapse	UE (upper arm)	Superficial	2.5	32
13	19	F	Primary	Groin	Superficial	3	55
14	68	M	Primary	Buttock	Deep	6.2	38
15	74	F	Primary	LE (lower leg)	Deep	9.5	54
16	33	M	Primary	LE (thigh)	Deep	2.8	73
17	26	F	Primary	Axilla	Deep	7	181
18	86	F	Primary	LE (thigh)	Deep	10	88
19	76	F	Primary	Buttock	Superficial	6.2	4
20	83	F	Primary	Buttock	Superficial	5.4	58
21	38	F	Primary	Tongue	Superficial	1.5	93
22	47	M	Primary	Back	Deep	4	11
23	28	M	Primary	Neck	Deep	14	121
24	22	F	Primary	Neck	Deep	7.2	75

LE: lower extremities, UE: upper extremities, F/U: follow-up.

**Table 2 cancers-15-02314-t002:** Treatment and outcome of initial tumor.

No.	Treatment	Margin Status	Local Relapse	Metastases	Outcome
1	Wide excision	R0	-	-	CDF
2	Wide excision	R0	-	-	CDF
3	Intralesional excision	R2	-	-	AWD
4	Wide excision	R0	-	-	CDF
5	Marginal excision	R1	-	-	CDF
6	Wide excision	R0	+	-	NED
7	Wide excision	R0	-	-	CDF
8	Wide excision	R0	-	-	CDF
9	Wide excision	R0	-	-	CDF
10	Marginal excision + adjuvant RT (50 Gy)	R1	-	-	CDF
11	Wide excision + adjuvant RT (48 Gy)	R1	-	-	CDF
12	Wide excision	R1	-	-	NED
13	Wide excision	R0	-	-	CDF
14	Wide excision	R0	-	-	CDF
15	Wide excision + adjuvant RT (60 Gy)	R0	+	-	AWD
16	Wide excision	R0	-	-	CDF
17	Marginal excision	R1	+	-	NED
18	Wide excision + adjuvant RT (66 Gy)	R1	-	-	CDF
19	Wide excision	R0	-	-	CDF
20	Marginal excision	R1	+	+	AWD
21	Wide excision	R0	-	-	CDF
22	Wide excision	R0	-	-	CDF
23	IMRT (60 Gy)	No surgery	-	-	CDF
24	IMRT (60 Gy)	No surgery	+	-	AWD

IMRT: intensity-modulated radiation therapy, CDF: completely disease-free, NED: no evidence of disease, AWD: alive with disease, DOD: dead of disease.

**Table 3 cancers-15-02314-t003:** Risk factors for local relapse.

Variables	*n*	5-Year Survival (%)	*p*-Value
Age (years)			
45>	12	83.3	0.52
45≤	12	63.4	
Sex			
male	10	83.3	0.25
female	14	68.4	
Primary site			
trunk	10	57.1	0.68
head and neck	5	80	
LE	6	83.3	
UE	3	100	
Localization			
deep	13	63.5	0.19
superficial	11	88.9	
Maximum diameter (cm)			
5>	15	90.9	<0.01
5≤	9	50.0	
Treatment			
surgery	18	79	0.58
Surgery + adjuvant RT	4	75	
RT	2	50	
Surgical methods (only surgical cases)			
wide excision	17	84.4	0.16
marginal + intra excision	5	60.0	
Margin status (only surgical cases)			
R0	14	82.1	0.49
R1 + R2	8	72.9	
Ki-67 (%)			
10>	12	71.4	0.78
10≤	12	79.5	

LE: lower extremities, UE: upper extremities.

**Table 4 cancers-15-02314-t004:** Treatment for local tumor relapse.

No. (Refer to Table 1)	Time to Local Relapse (m)	Treatment	Margin Status	Best Overall Response	Re-Local Relapse	F/U Periods after Treatment (m)
6	53	Surgery	R0	N/A	-	108
15	8	RT (60 Gy)	No surgery	PR	-	46
17	6	Carbon ion RT (70.4 Gy)	No surgery	SD	-	175
20	36	Surgery	R0	N/A	-	22
24	31	IMRT (60 Gy)	No surgery	PR	-	43

IMRT: intensity-modulated radiation therapy, N/A: not available, PR: partial response, SD: stable disease, F/U: follow-up.

**Table 5 cancers-15-02314-t005:** Summary of local relapse and distant metastases of LGMS in previous reports and this study.

Authors, Year of Publication	*n*	Local Relapse					Distant Metastases
		Simple Excision	Wide Excision	Excision + RT	Excision + Chemotherapy	Radical RT	
Mentzel et al., 1998 [1]	11	28.6% (2/7)	0% (0/1)	0% (0/3)	N/A	N/A	9.1% (1/11)
Montgomery et al., 2001 [3]	13	66.7% (6/9)	50% (1/2)	0% (0/2)	N/A	N/A	7.7% (1/13)
Meng et al., 2007 [4]	14	12.5% (1/8)		50% (1/2)	75% (3/4)	N/A	0% (0/14)
Kim et al., 2021 [5]	15	100% (2/2)	0% (0/13)	N/A	N/A	N/A	0% (0/15)
This study	24	50% (2/4)	7.1% (1/14)	25% (1/4)	N/A	50% (1/2)	4.2% (1/24)

RT: radiotherapy, N/A: not available.

## Data Availability

No new data were created or analyzed in this study. Data sharing is not applicable to this article.

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
