# Peer review of "Clinical Outcome of Low-Grade Myofibroblastic Sarcoma in Japan: A Multicenter Study from the Japanese Musculoskeletal Oncology Group"

_cancers, 2023, doi:10.3390/cancers15082314_

Round 1
Reviewer 1 Report
This research is about a rare disease in Japan, the authors presented the clinical outcome of Low-Grade Myofibroblastic Sarcoma.
Since this is a rare disease, not all the physicians have a chance to see the histology pictures, the authors have showed some histology result in the paper, this reviewer ask the authors to added some immunohistochemistry photographs to this paper.
Reviewer 2 Report
In this retrospective multicenter study the Authors aimed to analyze the clinical features and prognosis of 25 patients diagnosed with Low-grade myofibroblastic sarcoma.
The topic is interesting and the study well designed.
Introduction: please add some details about multimodal treatment and what is already known about prognosis of this sarcoma.
In order to homogenize data, I would exclude the only one case arising from bone.
3.3. it is not clear whether surgical excision was planned intralesional in one case. Table 2. case 11 wide excision but R1 margins...please check.case 12. marginal excision R0.
I would add a table including review of already described cases in the Literature.
Please have the paper checked by an English native speaker.
Round 2
Reviewer 2 Report
Thank you to the Authors for the efforts in ameliorating their paper.
In my opinion, it is now suitable for publication.